# Whole-Genome and Poly(A)+Transcriptome Analysis of the *Drosophila* Mutant *agn^ts3^* with Cognitive Dysfunctions

**DOI:** 10.3390/ijms25189891

**Published:** 2024-09-13

**Authors:** Aleksandr V. Zhuravlev, Dmitrii E. Polev, Anna V. Medvedeva, Elena V. Savvateeva-Popova

**Affiliations:** 1Pavlov Institute of Physiology, Russian Academy of Sciences, 199034 Saint Petersburg, Russiaesavvateeva@mail.ru (E.V.S.-P.); 2Saint-Petersburg Pasteur Institute, 197101 Saint Petersburg, Russia

**Keywords:** *Drosophila*, whole-genome sequencing, transcriptome sequencing, heat shock, hypomagnetic conditions, *prosalpha1*, learning and memory

## Abstract

The temperature-sensitive *Drosophila* mutant *agn^ts3^* exhibits the restoration of learning defects both after heat shock (HS) and under hypomagnetic conditions (HMC). Previously, *agn^ts3^* was shown to have an increased level of LIM kinase 1 (LIMK1). However, its *limk1* sequence did not significantly differ from that of the wild-type strain *Canton-S* (*CS*). Here, we performed whole-genome and poly(A)-enriched transcriptome sequencing of *CS* and *agn^ts3^* males normally, after HMC, and after HS. Several high-effect *agn^ts3^*-specific mutations were identified, including *MED23* (regulation of HS-dependent transcription) and *Spn42De*, the human orthologs of which are associated with intellectual disorders. Pronounced interstrain differences between the transcription profiles were revealed. Mainly, they included the genes of defense and stress response, long non-coding RNAs, and transposons. After HS, the differences between the transcriptomes became less pronounced. In *agn^ts3^*, *prosalpha1* was the only gene whose expression changed after both HS and HMC. The normal downregulation of *prosalpha1* and *Spn42De* in *agn^ts3^* was confirmed by RT-PCR. Analysis of *limk1* expression did not reveal any interstrain differences or changes after stress. Thus, behavioral differences between CS and *agn^ts3^* both under normal and stressed conditions are not due to differences in *limk1* transcription. Instead, *MED23*, *Spn42De,* and *prosalpha1* are more likely to contribute to the *agn^ts3^* phenotype.

## 1. Introduction

The study of the molecular genetic mechanisms of learning and memory started when mutations in several genes were shown to affect the behavior of *Drosophila melanogaster* [1]. Many of these mutations appeared to regulate cAMP-dependent signaling pathways. The first discovered was *dunce*. It impairs olfactory learning with negative reinforcement [2,3,4]. Different genes are responsible for different types of memories in the fruit fly; while short-term memory (STM) does not require protein synthesis, long-term memory (LTM) depends on de novo protein synthesis [5]. About 75% of genes responsible for human diseases, including neurological and neurodegenerative disorders, have structural analogues in *Drosophila* [6]. Since the molecular mechanisms of learning and memory are quite similar for vertebrates and invertebrates [7], the fruit fly is widely used as a model object in neurobiological research. Various biocollections have been created, for example TRiP, where fly strains carry interfering RNAs that specifically suppress the functions of certain genes activated by tissue-specific drivers [8]. However, there are several *Drosophila* mutants that have strongly manifested behavioral phenotypes, but their genetic basis is still obscure. One of such mutants is *Drosophila agn^ts3^*. It was isolated on the *Canton-S* (*CS*) background as a temperature-sensitive (*ts*) mutant with defects in cAMP metabolism and *ts*-dependent lethality in ontogenesis [9,10]. The mutation is located in the A/T-rich *agnostic* locus (X:11AB), harboring several other mutations affecting cAMP metabolism and chromosome recombination [11]. *agn^ts3^* shows a high frequency of chromosomal breaks and ectopic contacts, as well as under replication of salivary gland chromosomes [12,13,14,15]. Adult *agn^ts3^* males are defective in courtship learning at normal temperature, exhibit high levels of amyloid-like (Congo Red-positive) inclusions, as well as elevated brain levels of LIM-kinase 1 (LIMK1) and its product p-cofilin. After HS, the ability of *agn^ts3^* to learn and the levels of LIMK1 and p-cofilin returned to the level of the wild-type strain *CS* [16]. Using deletion mapping, restriction mapping, and radionuclide labeling, the *agn^ts3^* mutation was localized to the *limk1* gene (X:11B2) and was considered as a model for Williams–Beuren syndrome with cognitive impairments [11,17].

HMC is a relatively-little-studied type of stress factor that has multiple diverse effects on living systems [18]. Having evolved under a relatively stable geomagnetic background (25–65 µT), living organisms have developed adaptations to it. Hence, its elimination (shielding), as a rule, has negative effects, which include reducing cognitive abilities in both flies and humans, affecting skeletal muscles and bones, reducing reproductive ability, causing tumor formation, disrupting oxidative phosphorylation in mitochondria, etc. Among the possible mechanisms of the HMC action on living systems is its effect on electron spins and radical pairs, influencing several metabolic reactions, as well as enzymatic and signaling systems, such as cryptochrome receptors. The effects of HMC are poorly reproducible and may be nonspecific. However, they are attracting more and more research attention.

The HMC effects can be studied in *Drosophila* using a cylindrical chamber with a winding made of shielding soft magnetic alloy, which reduces the magnetic field strength inside by ~30–40 times. Like HS, a 12 h HMC restored courtship learning and memory in *agn^ts3^*, but suppressed 3 h memory in *CS* [19,20]. HMC significantly affected the levels of some proteins. In *CS*, it increased fat body protein (Fbp2) and decreased heat shock protein HSP27. In *agn^ts3^*, it induced alcohol dehydrogenase (Adh) and polo-like kinase (Plk4/Sak). HMC also significantly reduced the level of HSP70 in both strains, indicating that the molecular mechanisms of response to HMC are different from those of HS [21,22]. Of particular interest is the mechanism of learning recovery in *agn^ts3^* after both types of stress.

Despite extensive studies of the *agn^ts3^* phenotypic manifestations, their genetic basis remains unknown. Using the Sanger method, the *limk1* gene was sequenced for *agn^ts3^*, *CS*, and two other wild-type stains, *Oregon-R* and *Berlin* [23]. Various single nucleotide polymorphisms (SNPs) were observed in the *limk1* sequence of all the studied strains. Some of them were predicted to cause amino acid changes, but they did not impair the catalytic site of LIMK1 and possibly its biological activity. The only feature specific to *agn^ts3^* was the insertion of a 1734 bp long transposable S-element downstream of *limk1*, which, hypothetically, can affect *limk1* expression. The overall level of miRNA expression was lower in *agn^ts3^* than in *CS* and *Oregon-R*, both under normal conditions and after HS. At the same time, HS significantly increased the total level of miRNAs for each strain, with the greatest effect on *agn^ts3^*. Pronounced changes in the expression profile of *agn^ts3^* miRNAs may indicate an impairment of some systemic regulatory processes in this mutant, affecting the synthesis of a wide range of RNA products.

In addition to the S-element insertion, the *agn^ts3^* strain may carry some other mutations that act independently or in conjunction with rearrangements at the *agnostic* locus. In this study, we performed whole-genome sequencing of *agn^ts3^* and *CS*, as well as sequencing of their mRNA-enriched transcriptomes under normal conditions, after HMC, and after HS. Particular attention was paid to genes whose expression was affected by both types of stress, as they likely regulate *agn^ts3^* learning recovery.

## 2. Results

### 2.1. Whole-Genome Sequencing of CS and agn^ts3^

For the *CS* genome, the total number of observed variants (VN) that distinguished it from the reference genome was ~764 kb, with an average variant rate (VR) of one change per 186 bases. For the *agn^ts3^* genome, the corresponding values were ~663 kb and 1/213, respectively. Excluding variants common to both strains, the *CS*-specific VN and VR were ~259.5 kb and 1/543, respectively. The true values should be higher, as a significant share of bases with low sequencing quality was not taken into account (see Section 4). These values are of the same order as in [24]. When analyzing the nucleotide changes that may influence the *agn^ts3^* phenotype, we focused on the mutations with the highest predicted effect that were present exclusively in the mutant strain (Table 1). We also hypothesized that the functions of genes carrying a high number of such mutations (more than one) would be more likely to be affected. All identified sequence variants are presented in Archive_S1_genome_sequences_annotation.

In *CS*, our analysis method identified 718 affected genes (including all RNA isoforms) and 418 strain-specific affected genes carrying mutations of the High group. We limited our report to strain-specific mutations with a total number of ≥2 per gene. The resulting group included 12 genes. Among them, four (*Pdfr*, *cpo*, *nonA*, and *tmod*) are responsible for synaptic, neurological, and behavioral processes; one (*Tie*) regulates cell migration and survival; and seven have unknown biological functions (Table 2). We also noted genes belonging to the other three groups (Moderate, Low, and Modifier) with the highest number of changes per gene. Among them, *Strn-Mlck* likely regulates cytoskeletal dynamics in muscle; *shot* binds actin and microtubules, regulating axon extension and neuronal processes; and *dpr6* is involved in the sensory perception of chemical stimuli. It should be emphasized that even high-impact mutations do not necessarily lead to significant phenotypic changes since their impaired functions can be compensated by some other genes. However, the observed mutations may influence the behavioral and development processes in *CS*, distinguishing it from both *agn^ts3^* and the genomic reference strain.

In *agn^ts3^*, we observed 656 affected genes and 289 strain-specific affected genes carrying mutations of the High group. Nine of them are noted in Table 1 and Table 2. Among them, two genes have unknown functions, two (*Sdk* and *shakB*) regulate synaptic processes, one (*rdo*) is involved in the development of ocelli, and three (*Spn42De*, *Ssl2*, and *mthl3*) are predicted to regulate proteolysis, biosynthetic processes, and G protein signaling. *MED23*, the High group gene with the highest number of changes, encodes a component of the Mediator complex, a coactivator of RNA polymerase II, required for HS-dependent transcription. The orthologs of *MED23* and *Spn42De* in humans are responsible for intellectual developmental disorder 18 and prion disease, respectively. Only *sdk* and *shakB* are located on the X chromosome, both of which are far from the *agnostic* locus. As for the other three impact groups of *agn^ts3^*-specific mutated genes, *CG31817* has unknown functions, *sls* encodes a muscle protein involved in locomotion, and *sick* positively regulates F-actin-mediated axonal growth. None of them is located on the X chromosome.

None of the High group genes were localized at the *agnostic* locus (X:11AB). *limk1* carried several *agn^ts3^*-specific SNPs of the Low and Modifier groups (2 and 6, respectively). This was significantly lower compared to *CS* (29 and 23, respectively). *CS* also carried seven missense SNPs and an indel, resulting in the insertion of five amino acids; five of the above SNPs and the indel were reported in [23]. For the genes located in the X:11AB area, *CS* had only one gene (*ade5*) with the High impact change, while *agn^ts3^* did not have any. Thus, whole-genome sequencing did not reveal changes (SNPs or indels) with a significant effect on *agn^ts3^, limk1,* or any other gene at the *agnostic* locus. Instead, we identified other likely candidates for the molecular basis of the *agn^ts3^* phenotype, such as *MED23* and *Spn42De*.

### 2.2. Transcription Profiles of CS and agn^ts3^ under Normal Conditions and after Stress

Analysis of poly(A)-enriched transcriptomes of *CS* and *agn^ts3^* using iDEP revealed significant interstrain differences (Figure 1). Genes with the highest values of expression changes are shown in Appendix A. Data on all detected differentially expressed genes (DEGs) and pathways enrichment are presented in Archive_S2_transcriptome_sequences_annotation. A complete list of genes from the groups shown in Figure 1 is given in Appendix A.

Under normal conditions, our analysis method identified 138 differentially expressed genes (DEGs) in *agn^ts3^* that had an increased transcription levels compared to *CS*. About half of them were upregulated under all conditions, including *PGRP-SC*1b (the gene activating Toll signaling cascade), some long non-coding RNA (lncRNA) genes (*CR34335*), and several families of transposable elements (TEs), such as *copia*. These genes showed the highest level of activation in *agn^ts3^*. Other consistently upregulated genes included *grim* (apoptosis regulation in the central nervous system), *Lcp2* (cuticle component), *MST57DA* (male-specific peptide transferred to females during mating), *CP7FB* (oogenesis), *PGRP-SC2,* and *Cry* (blue-light-dependent regulator of circadian rhythms). The biological and molecular functions of many upregulated DEGs are unknown.

The number of DEGs upregulated in *agn^ts3^* slightly increased after a 12 h exposure to HMC, whereas HS decreased their number by 37. The number of genes exclusively upregulated under normal conditions mainly included various TEs. Some of the genes with known functions that were exclusively upregulated after HMC are *sisA* (developmental process), *Def* (activity against Gram-positive bacteria), and *Hsp70Bb* (reaction to hypoxia and HS). The group of genes exclusively upregulated after HS was specifically enriched in *Dm88* TEs, also known as *copia2.* Therefore, HS appears to activate this TE family in *agn^ts3^*.

A total of 155 genes showed decreased transcription levels in *agn^ts3^* compared to *CS*. DEGs downregulated under all conditions included *sordd2* (protein ubiquitination), *TotM* of the Turandot group encoding poorly characterized secreted peptides, *LysC* (defense protein), and *Spn42De*, which lacks a start codon according to *agn^ts3^* genome analysis. Genes specifically downregulated under normal conditions included *prosalpha1* (proteasome α1 subunit), whose level was dramatically decreased in *agn^ts3^*; *MESK4* (unknown functions); *TotZ*; *TotB*; *Lip* (lipid catabolic process); *MtnD;* and *MtnC* (response to copper ions). Again, HMC slightly increased the number of DEGs, while HS decreased it significantly. DEGs that were specifically downregulated after HMC included *IR7C* (detection of chemical stimulus), *NimC4* (apoptosis regulation), and *SR-CIV* (nervous system development). All the above groups also included TEs, lncRNAs genes, and genes with unknown functions.

KEGG pathway enrichment analysis identified several groups of DEGs that were upregulated in *agn^ts3^* under normal conditions (Appendix A). They included genes for the Toll and Imd (NF-κB) pathways, namely the genes of the PGRP family, which regulate antimicrobial and immune responses. Downregulated genes were involved in glycolysis/gluconeogenesis, galactose metabolism, and lysosome pathways. Similar enriched KEGG pathways were observed after HMC. Here, *Hex-C*, the gene for the key enzyme of glycolysis, was downregulated as well. HS eliminated the *agn^ts3^*-specific downregulation of genes involved in carbohydrate metabolism. Network analysis revealed an *agn^ts3^*-specific downregulation of genes involved in stress response and proteolysis. The above effect was observed only under normal conditions and not after stress.

The GO Biological Process enrichment analysis also revealed that many upregulated DEGs were involved in peptidoglycan metabolism and defense response to bacteria (Figure 2). They mainly included the genes of the PGRP family, as well as the genes for antibacterial proteins, such as listericin and attacin-A. UV- and heat-responsive genes were downregulated in *agn^ts3^*, along with some bacteria-responsive genes, mainly belonging to the Turandot group. In general, *agn^ts3^* showed a decrease in the activity of genes regulating the response to various types of abiotic stress. Similar groups of DEGs were observed following the HMC treatment. Here, we showed an enrichment of the upregulated DEGs that negatively affect the production of antibacterial peptides, as well as an enrichment of the downregulated DEGs that regulate proteolysis and response to heat. After HS, we did not observe any enrichment for the downregulated DEGs, likely due to the HS-dependent activation of stress-responsive genes.

For each strain, the effect of HMC on gene transcription was quite small. There were no DEGs for *CS* after HMC compared to normal conditions. In *agn^ts3^*, HMC significantly activated *prosalpha1*, which was downregulated compared to *CS* under normal conditions; hence, there was no interstrain difference after geomagnetic field shielding (Appendix A). To relax the selection criteria for DEGs, we omitted the Wald test, as specified by default in the iDEP settings. Thereafter, *CS* showed an increase in the expression of two *Beagle* transposons and a decrease in the expression of seven genes. These included two TEs; two genes encoding the copper-dependent protein metallothionein (*MtnC* and *MtnD*); *phu*, which regulates the response to hypoxia and nicotine; and *iotaTry*, which encodes serine protease. No enrichment for DEGs and pathways was observed in *agn^ts3^* after HMC. Interestingly, we did not observe any DEGs for *agn^ts3^*, when the Wald test was omitted.

In *CS*, HS induced the expression of 66 genes. As expected, the greatest activation was shown for genes regulating protein folding, response to hypoxia, and HS, i.e., the genes of the *Hsp70* family: *Hsp68*, *Hsp22*, *Hsp26*, *Hsp23*, *Hsp27*, *Hsp67Bc*, *DNAJ-1*, and *Hsp83* (listed in descending order of activation). Increased expression was also shown for some TEs such as *mdg3*, *opus,* and *Dm88*. In *agn^ts3^*, HS upregulated mostly the same genes as in *CS* but with a different order of expression. The magnitude of activation of *prosalpha1* was comparable to that for genes of the *Hsp70* family. In both strains, HS activated *AOX2*, which regulates caffeine metabolism. In contrast to *CS*, genes upregulated by HS in *agn^ts3^* included lysosomal genes of the LMan family. Different upregulated HSPs participate in longevity regulation, protein processing in EPR, spliceosome processes, and endocytosis (KEGG data). They are also involved in chromosome puffing, protein folding and refolding, and response to heat (GO Biological Processes data). Surprisingly, only a few genes with unknown functions were downregulated after HS in both strains.

Along with the DEG analysis for genes, we performed the DEG analysis for individual transcripts or isoforms (see Appendix A). On average, the number of differentially expressed isoforms was 2–3 times higher. Some of them were isoforms of the same genes with changed activity (e.g., *prosalpha-1RB*, *sordd2-RA*); others belonged to the genes whose overall expression level did not show any difference. For example, in both strains, HMC changed the level of expression of several DEGs, belonging to different functional classes. In some cases, different isoforms of the same genes were up- or downregulated (e.g., *Act5C-RA* and *Act5C-RC*).

Comparing the data from genomic and transcriptomic analyses, we found that in *agn^ts3^,* one of the nine genes with *agn^ts3^*-specific High-impact mutations was suppressed: *Spn42De*, an ortholog of human genes implicated in prion and Alzheimer’s diseases. No DEGs were observed for the X:11AB area, including *limk1* and its isoforms.

As mentioned above, *agn^ts3^* and *CS* differed in the expression levels of specific TEs, classified according to [25] and FlyBase [26]. Under normal conditions, the highest number of upregulated TEs was shown for the *copia*, *stalker2,* and *flea* families of LTR retrotransposons, while the highest number of downregulated TEs was shown for the *HeT-A* family of LINE retrotransposons. Both HS and HMC reduced the number of upregulated *copia* elements in the mutant strain. Notably, in *agn^ts3^,* HS significantly increased the amount of downregulated *opus* and upregulated *Dm88*. In *CS*, HS increased the amount of upregulated *opus* and *mdg3*, whereas HMC only had a minor effect on the activity of TEs. Interstrain differences in the activity of TEs under normal conditions and after HS may contribute to their phenotype and response to stress.

To confirm the data on DEGs obtained by transcriptome sequencing, we performed RT-PCR analysis of their expression in *agn^ts3^* relative to *CS* under normal conditions. Due to the large number of DEGs, we limited our analysis to genes of interest that presumably play a specific role in the *agn^ts3^* phenotype. They included *prosalpha1*, which was the only gene affected by both types of stress, and *Spn42De*, which was mutated and showed almost no expression in *agn^ts3^*. We also assessed the expression of *Cry*, a key regulator of circadian rhythms and response to HMC, and *Hsp70Bbb*, which regulates the response to HS. According to transcriptomic analysis, both of the above-mentioned genes were upregulated in *agn^ts3^*.

For *Cry*, we observed upregulation of ~4–6 fold, depending on the primer pair (Figure 3). *Spn42De* expression is virtually absent in *agn^ts3^*, consistent with the transcriptome sequencing data. Expression of *prosalpha1* was reduced in *agn^ts3^*, but the interstrain difference was not as large as revealed by the transcriptome analysis. *Hsp70Bbb* showed a trend toward increased expression in *agn^ts3^*, but the difference was not statistically significant. Thus, RT-PCR generally confirmed results of transcriptomic analysis, although for some genes the interstrain difference was smaller. The discrepancies may be due to the fact that we used poly(A)-enriched RNA fraction for transcriptome sequencing and total RNA fraction for RT-PCR.

## 3. Discussion

*agn^ts3^* is a mutant *Drosophila* strain characterized by *ts*-dependent lethality during ontogenesis and impaired learning ability, which is restored after various types of stress. Despite evidence for the involvement of LIMK1 in these processes, the molecular nature of the *agn^ts3^* phenotype remained unclear. Here, we performed whole-genome and poly(A)-enriched transcriptome sequencing of *agn^ts3^* and the wild-type strain *CS*, which serves as a control in various behavioral tests performed on *agn^ts3^*. Multiple mutations with predicted high impact were found in the *CS* and *agn^ts3^* genomes, but none of them was observed in the *limk1* or X:11AB areas. We did not perform de novo genome sequencing. Hence, it was impossible for us to detect the large structural rearrangements in the *agn^ts3^* genome near *limk1*, as shown by restriction mapping [11], as well as the insertion of S-element downstream of *agn^ts3^ limk1* [23].

Recently, increased *limk1* activity was shown to impair short-term courtship memory in *Drosophila* [27]. Similarly, *agn^ts3^* is impaired in learning and memory under normal conditions and has increased levels of LIMK1 and its product p-cofilin compared to *CS* [16]. However, our study showed that the transcription level of *limk1* was the same in both strains and did not change after stress. Thus, although LIMK1 protein level may be altered in *agn^ts3^*, this is probably a result of the regulatory effects of some other genes affecting multiple biological processes in the mutant strain. Since the two lines showed significant differences both at the genome and transcriptome levels, it is quite difficult to identify the key mutation affecting the *agn^ts3^* phenotype. Instead, we can focus on the mutant genes and DEGs that are likely to contribute more than others. They may also be the subject of future research into the interaction of genes, stress, and learning abilities.

We have found several high-effect genes with *agn^ts3^*-specific mutations that likely influence its phenotype. Among them is *MED23*, a cofactor of RNA polymerase II required for the expression of HS-specific genes [28]. Hypothetically, this could explain the decrease in interstrain differences in mRNA profiles after heat shock. MED23 also participates in metabolic homeostasis by suppressing FOXO1, a key metabolic transcription factor required for gluconeogenesis [29]. Notably, genes regulating carbohydrate metabolism, including glycolysis and gluconeogenesis, were downregulated in *agn^ts3^* compared to *CS* under normal conditions and after HMC. Although the expression level of *MED23* was unchanged in *agn^ts3^*, the splice-site mutation and the appearance of a stop codon likely resulted in abnormal mRNA and protein products with loss of functional activity.

*Spn42De* is an ortholog of human genes, encoding serpins such as *SERPINI1,* whose expression is reduced in patients with Creutzfeldt–Jakob prion disease [30]. Loss of the start codon in *agn^ts3^ Spn42De* should block its transcription, which was confirmed by both transcriptome sequencing and RT-PCR. Amyloid-like inclusions accompany the development of multiple neuropathologies. In *agn^ts3^*, such inclusions are observed at normal temperature and disappear after HS [16], possibly due to the activation of the HSP system.

The *Drosophila* immune system includes several innate molecular systems that sense infection and produce effector molecules. Most of them are antimicrobial peptides [31]. Gram-positive bacteria and fungi activate the transmembrane receptor Toll, regulating the production of drosomycin. The Imd pathway controls defense against Gram-negative bacteria. Both biological processes seem to be activated in *agn^ts3^*. Therefore, this strain may be infected with some microorganisms. *Drosophila* memory is negatively affected when the immune system is stimulated by the activation of peptidoglycan receptor protein (PGRP) [32], similar to what was observed in our case. However, this does not explain why learning and memory are restored after HMC and HS, although the level of *PGRP* expression remains high.

Among the genes constantly upregulated in *agn^ts3^* is *Cry*, which encodes cryptochrome (CRY), a blue-light-sensitive protein. CRY serves as a negative regulator in the molecular circadian clocks, repressing the CLOCK–CYCLE transcription complex [33]. Circadian rhythms regulate memory consolidation in *Drosophila* by influencing neurons of the mushroom body and the central complex [34,35]. Thus, learning impairment in *agn^ts3^* may partly result from hyperactivation of CRY-dependent pathways.

Genes of the Turandot group were specifically downregulated in the mutant strain. The above genes are induced under stress conditions [36]. For example, in *CS* females, they were induced after daily repeated HS [37]. HS reduced the number of interstrain DEGs as well as the interstrain behavioral differences [16]. Surprisingly, only a few genes with unknown functions were downregulated after HS in each strain. Various types of stress, including HS, have been shown to downregulate a conservative group of genes that is primarily associated with metabolism [38]. It should be noted, however, that in our study, HS was quite mild—a single application of 37 °C for 30 min, followed by 1 h at room temperature.

HMC is known to cause various behavioral abnormalities. For example, in mice, it increases anxiety-like behavior [39] and leads to impairments in adult hippocampal neurogenesis and cognition [40]. The latter is due to a decrease in the content of reactive oxygen species (ROS). Transcriptome sequencing showed that 72 h HMC (~190 times shielding) upregulated genes that negatively regulate protein metabolic processes and cell proliferation, while genes that respond to hypoxia and ROS were downregulated. In human neuroblastoma cells, 2-day application of HMC upregulated about two hundred genes and downregulated ten times more genes, including *CRY2* and *MAPK1* [41].

In our study, HMC did not affect any of the above genes when comparing expression after HMC and normal expression. The low effect of HMC on transcriptomes in our case may be due to the limited time or extent of HMC application, which was insufficient to influence gene transcription, despite its pronounced effects on fly behavior [20]. Moreover, we did not observe any changes in the transcription of genes and RNA isoforms of *Drosophila* proteins, the level of which was previously reported to change after HMC [21]. This may indicate a stronger effect of HMC on protein translation.

Among the DEGs that may play a significant role in the response to HMC were genes involved in copper binding. Metallothioneins play an important role in protecting the body from oxidative stress. Their expression is induced by heavy metals via metal-responsive transcription factor 1 (MTF-1). MtnA—MtnD protect the fly from Cu and Cd, ensuring redox balance [42]. MTF-1 is an evolutionarily conserved protein that regulates the response to various types of stress, including oxidative stress and hypoxia [43]. Chemical compounds containing electron spins or radical pairs likely mediate the effects of HMC on living systems [18]. Thus, the downregulation of *MtnC* and *MtnD* by HMC may be caused by spin changes in transition metals such as Cu and Cd, which affect their binding to MTF-1. *MtnC* and *MtnD* were also downregulated in *agn^ts3^* compared to *CS* under all conditions, possibly making this strain more vulnerable to oxidative stress.

The only gene with a reduced expression in *agn^ts3^* compared to *CS* under normal conditions is *prosalpha1,* which was normalized after both HMC and HS. In *Drosophila*, *prosalpha1* encodes the α1 subunit of the proteasome, expressed in the circulatory system and involved in ubiquitin-dependent protein catabolism. Little is known about its biological functions. According to FlyBase, mutations in *prosalpha1* may cause abnormal neuroanatomy and lethality. It is orthologous to the human gene encoding the 20S proteasome alpha subunit. SNPs in the above gene are associated with diabetes, myocardial infarction, coronary artery disease, and end-stage kidney disease [44]. Genome sequencing data showed that *agn^ts3^* carries only two SNPs of the Modifier type downstream of the *prosalpha1* gene. Similar SNPs are also present in the wild type. Probably, the observed changes in the activity of *agn^ts3^ prosalpha1* depend on some other mutant genes.

In addition to the expression of protein-encoding genes, the fly strains studied showed significant differences in the activity of lncRNAs and TEs. lncRNAs participate in DNA methylation, histone modifications, chromatin remodeling, and transcriptional regulation [45]. They also play a role in the formation of chromatin compartments and the spatial organization of the cell nucleus [46,47]. The organization of *agn^ts3^* heterochromatin significantly differs from that of *CS* [16], which may be caused by some lncRNAs and different distribution or activity of some TEs.

About 20% of the *D. melanogaster* genome consists of TEs (34,805 copies), predominantly retrotransposons of the LTR and LINE orders, which make up 12% and 5% of the genome, respectively [48]. This is consistent with our data for DEGs comprising mainly LTR TEs. However, in our case, *copia* turned out to be the superfamily with the highest number of DEGs, and *gypsy* included only a few DEGs, as opposed to their copy number in the genome: >11,000 for *gypsy* and ~1000 for *copia* [48]. Since we did not perform de novo genome sequencing, we were unable to determine the exact number and location of TEs in both strains. The increase in transcript level may be caused by an increase in copy number and/or transposing activity of TEs. An inbred *Drosophila* strain showed an increase in the transposition of *copia* elements over several generations without a drastic change in the other three TEs [49]. This may explain the *agn^ts3^*-specific increase in *copia* expression, both under normal conditions and after stress.

In our study, HS activated mainly the TEs of *opus*, *mdg3,* and *Dm88/copia2* families in *CS* and the TEs of *Dm88/copia2* family in *agn^ts3^*. In the *D. melanogaster* genome, there is a TE cluster located between *Hsp70Ba* and *Hsp70Bbb* (3R:12,468–12,503 kb). It contains 36 transposons, including 16 *Dm88* elements, as well as *lncRNA CR32865*. The latter is a chimeric product of *Dm88* and the *Invader1* sequence, which itself is a fragment of the *Hsp70* promoter [50,51]. Under normal conditions, no interstrain differences in expression of TEs were observed. After HS, *agn^ts3^* showed a significant increase in the transcriptional activity of eleven cluster elements, including eight *Dm88.* In *CS*, HS increased the activity of only four *Dm88* and to a lesser extent than in *agn^ts3^*. HS-dependent upregulation of *Dm88/copia2* and *Invader1* of the same TEs cluster was also shown by other researchers [52]. The difference between *CS* and *agn^ts3^* in response to HS may be related to higher activation of the Hsp70 transposon cluster in the mutant strain.

In summary, we observed significant differences between *CS* and *agn^ts3^* at both the genome and transcriptome levels. The lack of interstrain differences in *limk1* sequence and transcription level suggests that the previously observed HS-dependent changes in *agn^ts3^* LIMK1/ p-cofilin levels may be caused by other mutations affecting response to HS, such as *MED23*. *Cry* was significantly upregulated in *agn^ts3^*, which may affect its sensitivity to light and HMC. The activity of *prosalpha1* was low in *agn^ts3^* and increased after both HMC and HS. This reveals a possible role for *prosalpha1* in learning and stress response. Interstrain differences in the transcriptional activity of TEs may result from spontaneous activation of *copia/copia*-like transpositions in *agn^ts3^* during the long time of maintenance of this strain. Further studies are needed to clarify the role of the identified mutations in the learning processes of flies under various conditions.

## 4. Materials and Methods

### 4.1. Fly Strains

Fly strains were taken from the Biocollection of Pavlov Institute of Physiology RAS for the Study of Integrative Mechanisms of Nervous and Visceral Systems, Saint Petersburg, Russia. The wild-type strain *Canton-S (CS)* was obtained from Bloomington Drosophila Stock Center, USA. *agn^ts3^* was an EMC-induced *ts* mutation selected from a *CS* background in 1981 and bred in Biocollection since that time. It should be noted that BDSC *CS* and *CS* used for *agn^ts3^* selection represent different populations of flies, so significant differences in their genetic backgrounds are expected. The *agn^ts3^* phenotype has been tested periodically, being checked for developmental lethality at 29 °C and for learning disability in conditioned courtship suppressing paradigm [53]. Flies were reared at 22 ± 0.5 °C on standard yeast–raisin medium with 8 a.m.–8 p.m. daily illumination. Before experiment, adult males were kept for 4–5 days at 25 ± 0.5 °C. Nucleic acids were isolated from flies in the first half of the day.

### 4.2. Hypomagnetic Conditioning

Hypomagnetic conditions (HMC) were applied as described in [19]. Briefly, 4-day-old males were placed overnight (12 h) in food-containing vials in a cylindrical chamber coated with AMAG-172 shielding material. The shielding factor in the chamber was 38. After exposure to HMC, the flies were kept for 1 h at 25 °C before experiment in normal terrestrial magnetic field.

### 4.3. Heat Shock Conditioning

Heat shock was applied to 5-day-old males according to [54]. Flies were placed in empty glass vials in a water bath for 30 min at 37 °C and then kept for 1 h at 25 °C in food-containing vials before the experiment.

### 4.4. DNA Extraction and Whole-Genome Sequencing

For each strain, DNA was purified from ten 5-day-old males using DNA extraction on a spin column (Extract DNA Blood BM013, Evrogen, Moscow, Russia), according to the manufacturer’s recommendations. The concentration of nucleic acids was measured using Eppendorf BioPhotometer (Hamburg, Germany). The quality of genomic DNA was assessed by electrophoresis in 1.5% agarose gel using Agagel Mini system (Biometra, Göttingen, Germany). DNA libraries for whole-genome sequencing were prepared using NadPrep EZ DNA Library Preparation Kit v2 (Nanodigmbio, Nanjing, China). Sequencing was performed using DNBSEQ-G50 High-throughput Sequencing Set (MGI, Shenzhen, China), obtaining 160 bp paired-end reads.

### 4.5. RNA Extraction and Transcriptome Sequencing

For each strain and condition (normal conditions, HMC, HS), three parallel independent RNA extractions were performed. RNA was extracted from ten 5-day-old males as follows: Flies were homogenized in 300 μL TRI reagent (Sigma-Aldrich, St. Louis, MO, USA, T9424) and centrifuged for 5 min at 12,000× *g*. The supernatant was mixed with an equal volume of 96% ethanol, loaded onto HiPure Total RNA Kit R401102 columns (Magen, Guangzhou, China), and purified according to the manufacturer’s protocol. The concentration of nucleic acids was measured using Eppendorf BioPhotometer. The quality of RNA was assessed by electrophoresis in 1.5% agarose gel using Agagel Mini system. A total of 200 ng total RNA was used for library preparation. VAHTS mRNA Capture Beads 2.0 N403 (Nanodigmbio) were used to enrich poly(A)-enriched RNA in RNA fraction. Libraries were prepared using MGIEasy RNA Library Prep Kit (MGI). Sequencing was performed using DNBSEQ-G50 High-throughput Sequencing Set (MGI), obtaining 160 bp paired-end reads.

### 4.6. Genome Assembly and Annotation

Assessment of raw read quality was performed using FastQC [55]. Low-quality and unpaired reads were removed using Trimmomatic [56], with the following filtration parameters: leading, 15; trailing, 15; sliding window, 4:22; and minlen, 36. Paired-end reads were aligned using BWA-MEM [57] to the primary assembly of the *Drosophila melanogaster* genome (version BDGP6.46) and analyzed using samtools/bcftools utilities [58]. The average read depth (DP) for *CS* and *agn^ts3^* was 42.9 and 28.6, respectively. Indels positions were normalized using bcftools, and indels were filtered (IndelGap = 10). In the final assembly, only positions with QUAL ≥ 20 and DP > 10 were considered. Hence, ~16.60 Mb (12.10%) for *CS* and ~21.83 Mb (15.87%) for *agn^ts3^* with low sequencing quality or low DP were excluded from the analysis. Mutations were annotated using SnpEff v4.3 [24], which predicts the strength of a mutation effect as follows: High (e.g., frameshift, mutation in splicing site, stop codon generation, start codon loss), Moderate (e.g., missense variant, inframe deletion or insertion), Low (e.g., synonymous variant, splice region variant), and Modifier (e.g., non-coding transcript variant, upstream or downstream variant).

### 4.7. Transcriptome Reads Assembly and Quantification

Counts of reads mapped to the *Drosophila* genome and transcriptome were obtained similar to [59]. Specifically, for each strain, pair-end reads were mapped to the primary assembly of the *Drosophila melanogaser* genome (version BDGP6.46), with Ensemble annotation using STAR v. 2.7.11a [60]. The values for minimum and maximum intron size were 44 and 70,000, respectively. STAR options “–outSAMtype BAM SortedByCoordinate–quantMode TranscriptomeSAM” were enabled to generate alignments to both genome and transcriptome. Quantitative analysis of gene and RNA isoform reads was performed using RSEM v1.3.3 [61] to obtain the expected read counts.

The exact set of commands used to assemble and annotate DNA and RNA sequences (excluding the iDEP analysis step described below) is provided in Appendix A. Data on the expected read counts that was processed by iDEP to obtain DEGs can be found in Archive_S3_expression_counts.

### 4.8. DEG Analysis

The expected read counts (three replicates for each strain) were analyzed using iDEP 2.01 online service [62,63]. *D. melanogaster* Ensembl annotation and read counts data parameters were selected. The DESeq2 method [64] was used to find differentially expressed genes (DEGs), with FDR cutoff = 0.1 and min fold change = 2. Additionally, the Wald test was used to limit the number of DEGs and increase the specificity of the analysis. In each case, the strains were compared in pairs. Venn diagram construction and enrichment analysis were performed for genes only (not for RNA isoforms). FlyBase IDs were used to automatically annotate *Drosophila* genes to detect enrichment of DEGs for different pathways (KEGG) [65] and processes (GO Biological Process) [66]. Pathway enrichment was revealed using the pre-ranked fast gene set enrichment analysis tool (preranked fgsea) [67]. Enrichment data were obtained for pathway networks (cutoff = 0.3).

### 4.9. Reverse Transcription and Semi-Quantitative Real-Time PCR

Five replicates were prepared independently for each strain. Total RNA was isolated as described in Section 4.5. RNA (1 μg) was treated with DNAse I (Servicebio, Wuhan, China, G3342) and reverse transcribed by MMLV reverse transcriptase (Evrogen, Moscow, Russia, #SK022S) according to the manufacturer’s protocol, using random primers and RNAse inhibitor (Syntol, Moscow, Russia, #E-055). PCR was performed on StepOnePlus (Applied Biosystems, Waltham, MA, USA) using qPCRmix HS SYBR+LowROX (Evrogen, #PK156L). The expression levels of *rpl32* and *EF1α2* served as internal controls. Relative expression levels were calculated using StepOne software v2.3 (Applied Biosystems). Primer sequences and PCR parameters are given in Appendix A. Files containing the detailed results of RT-PCR experiments are given in Archive_S4_RT-PCR_data.

## Figures and Tables

**Figure 1 ijms-25-09891-f001:**
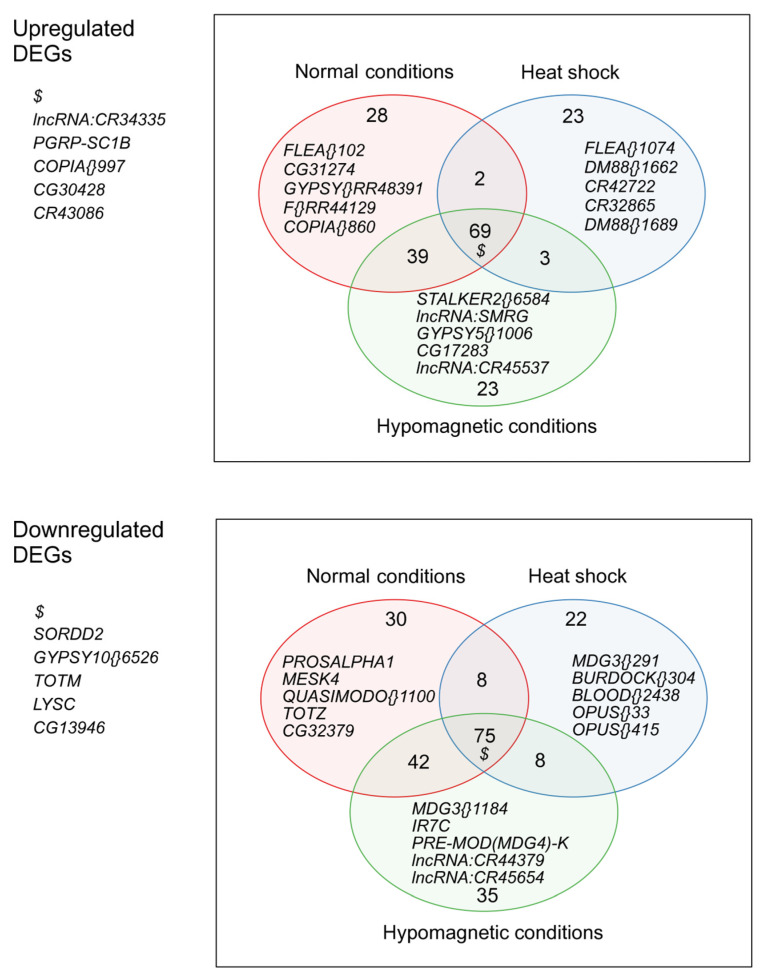
Differentially expressed genes in *agn^ts3^* relative to *Canton-S* (*CS*) under different conditions (Venn diagrams). The names of five genes showing the maximum change in expression under specific conditions are shown. $ refers to genes that showed the interstrain difference of expression across all conditions.

**Figure 2 ijms-25-09891-f002:**
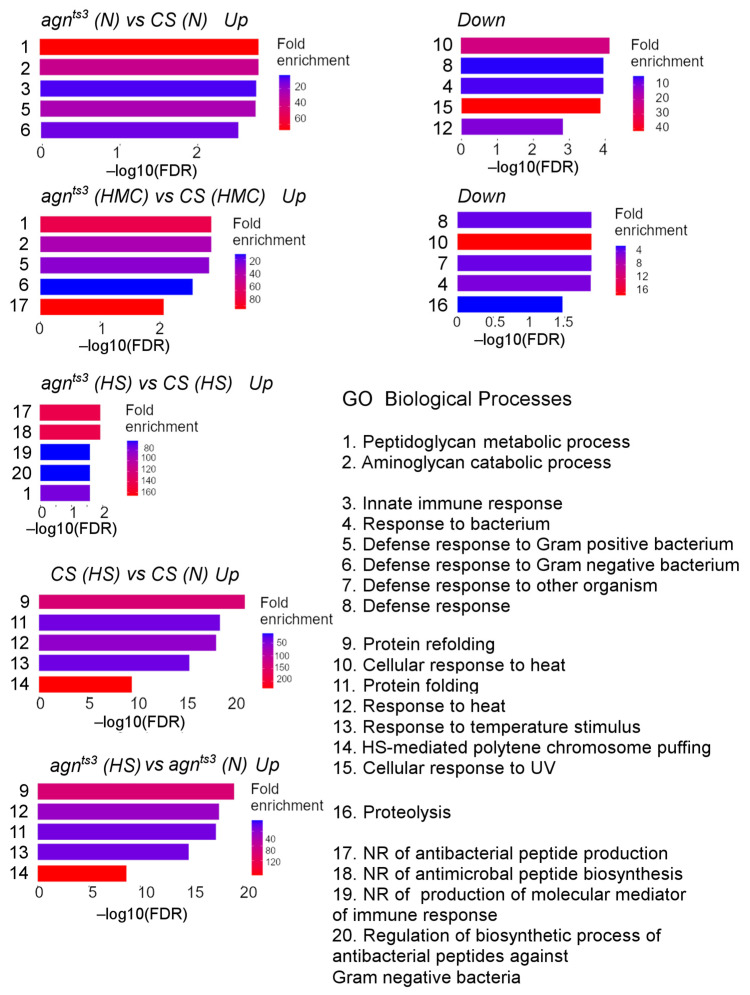
GO Biological Processes enrichment. Abbreviations: Down, downregulated; FDR, false discovery rate; HMC, hypomagnetic conditions; HS, heat shock; N, normal conditions; NR, negative regulation; Up, upregulated. Because there is significant overlap between the genes from the different processes, only five processes are shown for each case.

**Figure 3 ijms-25-09891-f003:**
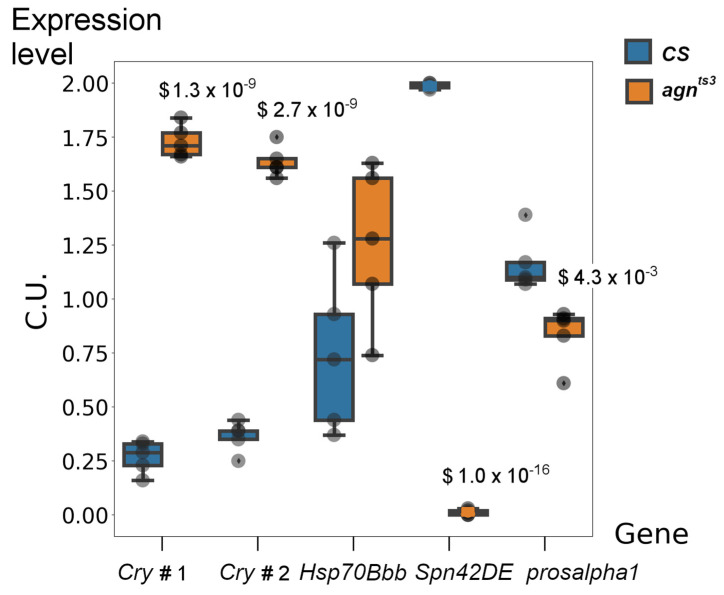
Normalized gene expression levels in *CS* and *agn^ts3^* (RT-PCR data). Expression levels are indicated in conditional units (C.U.). The average expression level of each gene was set to 1. Statistical differences: $ from *CS* (two-sided t-тест, n = 5, p is given above the box plot). *Cry* #1 and *Cry* #2 correspond to different pairs of primers. The median is shown as a dark line. Outliers are indicated by a diamond.

**Table 1 ijms-25-09891-t001:** Whole-genome sequences of *Canton-S* (*CS)* and *agn^ts3^*: the most important mutations.

Strain	High	Moderate	Low	Modifier
*CS* all	All: 785; 4: 1(*Tie*); 3: 8(*tun, CG9698, nonA CG18538, twin*)	All: 15,766; 109: 19(*Muc14A)*	All: 23,023; 290: 1(*shot*)	All: 33,986; 2025: 1(*dpr6*)
*agn^ts3^* all	All: 656; 3: 13(*CG18538*; *MED23*, *Tep5*, *Sdk*, *twin*)	All: 14,467; 87: 1(*sls*)	All: 21,984; 249: 2(*sls*)	All: 34,052; 1885: 1(*Ptp61F*)
*CS*- specific	All: 418; 4: 1(*Tie*); 3: 1(*CG9698*); 2: 18(*CG10183*, *CG13700*, *CG8136*, *CG9500*, *CR44091*, *CR45136*, *Pdfr*, *cpo*, *nonA*, *tmod*)	All: 10,193; 66: 3(*Strn-Mlck*)	All: 16,341; 170: 1(*shot*)	All: 32,791; 1254: 1(*dpr6*)
*agn^ts3^*- specific	All: 289; 3: 2(*MED23*, *Tep5*); 2: 5(*CG43829*, *Spn42De*, *Ssl2*, *mthl3*, *rd0*, *Sdk*, *shakB*)	All: 7,613; 69: 1(*Msp300*)	All: 13,205; 121: 1(*sls*)	All: 31,626; 875: 1(*bru1*)
*CS*/*agn^ts3^* common	All: 379; 3: 2(*CG18538*, *twin*); 2: 12(*AttC*, *CG14692*, *CG31268*, *CG43092*, *CG4580*, *Cp1*, *Cyp4p1*, *Cyp6a14*, *Ugt86D*, *inaD*, *tun*)	All: 10,562; 66: 1(*CG31817*)	All: 18,011: 128: 1(*sls*)	All: 32,723; 1240: 1(*sick*)

Abbreviations: High, Moderate, Low, Modifier—the predicted mutation impact. All—the total number of affected RNA isoforms for all mutated genes. Genes with the highest number of changes are shown. The data in the table are presented as x: y(z), where x is the number of changes in a single gene, y is the number of affected RNA isoforms, and z is the name of the gene.

**Table 2 ijms-25-09891-t002:** Position in the genome and function of strain-specific mutant genes with a predicted High impact.

Gene	Genome Positions	Mutation Effect	Gene Functions
*CS*-specific genes			
*Tie*	3L:4510698..4532145	start loss/frameshift	Predicted: cell survival and migration
*CG9698*	3R:30514777..30516980	stop lost	Unknown
*CG10183*	3R:23596561..23599034	stop gained/splicing defect	Unknown
*CG13700*	3L:18298742..18301260	stop gained/ splicing defect	Unknown
*CG8136*	3R:8721371..8723051	stop gained/ splicing defect	Unknown
*CG9500*	2L:6357005..6358318	frameshift/ splicing defect	Unknown
*CR440* *91*	3R:31680596..31681019	splicing defect	Unknown
*CR45136*	2R:10853546..10854495	splicing defect	Unknown
*Pdfr*	X:2552206..2578640	frameshift	Circadian processes, development of the flight motor system, regulation of mating
*cpo*	3R:17919832..18018892	frameshift/ splicing defect	Synaptic transmission, climate adaptation, olfaction
*nonA*	X:16361569..16371547	stop gained/frameshift	Visual behavior
*tmod*	3R:30532607..30578905	splicing defect	Notch signaling regulation, actin filament organization
*agn^ts3^*-specific genes			
*MED23*	2R:22888928..222894425	stop gained/ splicing defect	RNA polymerase II coactivator, transcription response to heat shock. The human ortholog(s) are involved in autosomal recessive intellectual developmental disorder 18.
*Tep5*	2L:19558916..19563049	frameshift	Unknown (pseudogene)
*CG43829*	2R:15193996..15194505	frameshift	Unknown
*Spn42De*	2R:6885552..6888262	start lost/ splicing defect	Predicted: negative regulation of proteolysis. The human ortholog(s) are involved in Alzheimer’s disease and familial encephalopathy.
*Ssl2*	3R:29030951..29032642	frameshift	Predicted: regulation of biosynthetic processes.
*mthl3*	2R:17448533..17452598	splicing defect	Predicted: G protein-coupled receptor signaling pathway, determination of adult lifespan
*rdo*	2L:18012380..18070246	frameshift/ stop gained	Ocelli development
*sdk*	X:686834..749496	stop gained	Actin branching, epithelial remodeling, synapse formation
*shakB*	X:20761071..20927050	frameshift/ splicing defect	Structural component of the gap junctions at electrical synapses

Genome positions and gene functions are given according to FlyBase.

## Data Availability

The original sequencing data presented in this study are openly available in NCBI, Sequence Read Archive (for genome analysis, raw reads: PRJNA1129042; for transcriptome analysis, raw reads: PRJNA1134087; reads aligned to the genome: PRJNA1152604).

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
