# Peer review of "Whole-Genome and Poly(A)+Transcriptome Analysis of the Drosophila Mutant agnts3 with Cognitive Dysfunctions"

_ijms, 2024, doi:10.3390/ijms25189891_

Round 1
Reviewer 1 Report
Comments and Suggestions for Authors
In their submitted manuscript, Aleksandr et al. performed whole-genome and 14 poly(A)+transcriptome sequencing for CS and agnts3 males and found that the pronounced interstrain differences in transcription profiles, mainly including genes of defense response, proteolysis, long non-coding RNAs and transposable elements, which is interesting. To me, this work represents a step forward in the understanding of the temperature-sensitive Drosophila mutant agnts3 in learning and memory defects. I have the following questions/comments.
1. Figure 1. High-throughput sequencing often contains errors and mistakes, please use RT-qPCR to check the labeled genes in Volcano Plot.
2. The method section requires substantial improvement to enhance its clarity and reproducibility for other researchers.
3. Some modifications/corrections in the manuscript may be needed.
Comments on the Quality of English Language
n/a
Author Response
First of all, we would like to thank Reviewer for the valuable comments and suggestions. Here are our responses:
Comments 1: Figure 1. High-throughput sequencing often contains errors and mistakes, please use RT-qPCR to check the labeled genes in Volcano Plot.
Response 1: Because of the large number of DEGs, RT-PCR was used to confirm the expression changes only for several genes of interest, including prosalpha1, Spn42De, and some others. We also replaced Volcano Plot by Venn diagram (Page 6).
Comments 2: The method section requires substantial improvement to enhance its clarity and reproducibility for other researchers.
Response 2: We rewrote the Methods section to make it more clear. For example, we included some additional subchapters for the procedures of whole-genome sequencing, transcriptome sequencing, and DEG analysis (Page 13, 14). For genome and transcriptome analysis, the full procedure of reads assembly is described, including all the necessary commands and parameters (see Supplementary Materials, Text S2). We believe that сurrently all the methods are described in sufficient detail.
Comments 3: Some modifications/corrections in the manuscript may be needed.
Response 3: We revised the text of the manuscript, including Results and Discussion sections, to make it more logical and coherent.
Reviewer 2 Report
Comments and Suggestions for Authors
The subject of the research is interesting and potentially valuable. However, the english is so poor that it is difficult to follow the content. Sometimes it is impossible to understand the sentence at all. Besides, the structure of the story should be improved and simplified. Authors have to be clear what they compare and what conditions they try. The work and the abstract is inconclusive. Authors must work on it and figure out what data tell us after all. Otherwise do more research. If you have some hypothesis, you must seek for the experimental evidence supporting your ideas. The selection of figures and the tables content must be improved too. For instance Table 1 should show the mutations at the protein level and possible implications of it. Figure 1 is poor choice. Venn diagrams provide better summary on the found groups of DE genes. Figure 2 is too busy and not clear in its message. It needs further analysis and more clear summary of what is found, what is important for the story. Fig. 3 is not really informative, it can be compressed or replaced by some alternative illustration. What is the point of it? The discussion should be more connected to the results and do a clear emphasis on the most important findings in the manuscript. In addition the paper should conform open access policies, sequencing and expression data should be deposited in open access databases, like Arrayexpress or NCBI GEO. At present the paper is not yet at the condition of taking it with major revisions. It has to be thoroughly re-written, restructured, optimised and data become publicly available for the inspection and reanalysis.
Comments on the Quality of English LanguageEnglish is extremely poor. Authors have to recruit professional interpreters.
Author Response
First of all, we would like to thank Reviewer for the valuable comments and suggestions. Here are our responses:
Comments 1: The english is so poor that it is difficult to follow the content. Sometimes it is impossible to understand the sentence at all. Besides, the structure of the story should be improved and simplified.
Response 1: We revised the text, made it more simple and coherent, removed unnecessary citations, and corrected errors.
Comments 2: The work and the abstract is inconclusive. Authors must work on it and figure out what data tell us after all. Otherwise do more research. If you have some hypothesis, you must seek for the experimental evidence supporting your ideas.
Response 2: The main objective of the study was to compare two Drosophila strains at the genome and transcriptome level, under different conditions where these strains show learning differences. It was previously assumed that the strain agnts3 differs from the wild type strain Canton-S due to a mutation or transcriptional changes of the gene limk1. As we have shown here, there are no significant interstrain differences either in the structure of limk1, or in its transcription. At the same time, we revealed a set of mutant genes, as well as genes with changed expression in agnts3 , which probably сontribute to its learning ability. This is completely new information, which opens the way to further research. We tried to explain it more clearly in the text.
Comments 3: Table 1 should show the mutations at the protein level and possible implications of it.
Response 3: Table 1 shows the general information about the number and types of nucleotide changes in both strains. It was impossible to include the detailed information about each specific gene in it. Instead we included a new column in Table 2 (Page 4, 5), which describes a mutation effect for genes of interest with High impact, – e.g., stop codon generation, etc.
Comments 4: Figure 1 is poor choice. Venn diagrams provide better summary on the found groups of DE genes.
Response 4: We replaced Volcano Plots by Venn diagrams (Page 6). We also rewrote the Result section to make it more compact and more related to information given in Figure 1.
Comments 5: Figure 2 is too busy and not clear in its message. It needs further analysis and more clear summary of what is found, what is important for the story.
Response 5: We limited the number of processes for each case to five, which made Figure 2 more simpler and clearer (Page 8).
Comments 6: Fig. 3 is not really informative, it can be compressed or replaced by some alternative illustration. What is the point of it?
Response 6: The point of Figure 3 was to show differences of the expression for specific transposon families, between the two strains and under various conditions. For example, HS specifically induces the expression of Dm88. To simplify the manuscript, we moved this figure to Supplementary materials (Figure S1).
Comments 7: The discussion should be more connected to the results and do a clear emphasis on the most important findings in the manuscript.
Response 7: We rewrote the Discussion section, simplified it and made it closer to the results. We tried to reveal possible relationships between various facts discovered. The hypothetical mechanisms of agnts3 learning impairments and its recovery after stress are discussed in the light of these facts.
Comments 8: In addition the paper should conform open access policies, sequencing and expression data should be deposited in open access databases, like Arrayexpress or NCBI GEO.
Response 8: In the first version of the manuscript, we provided IDs (NCBI, Sequence Read Archive) to access raw sequencing data (Page 15, 582-583). We also provided data of analysis of genomes and transcriptomes (Supplementary Materials, Archive S1 – S3, Page 15, 563-565). In the revised version of the manuscript, according to Reviewer’s suggestion, we also added raw data of RT-PCR analysis to Supplementary Materials (Page 15, 566-567).
Round 2
Reviewer 1 Report
Comments and Suggestions for Authors
The authors have answered my questions. I don't have other questions.
Author Response
Comments 1: The authors have answered my questions. I don't have other questions.
Reply 1: Thank you very much for your comment.
Reviewer 2 Report
Comments and Suggestions for Authors
The paper is more readable than before, but some parts are still difficult to follow. I see that authors have taken suggestions and improved their tables and figures. However, headers to the tables need further improvements (table 1 for instance). Figure 3 is still too busy an barely informative. Try to focus on the major message and provide all details. Note, that GO titles say nothing about underlying genes and gene numbers. It is important to show the gene names as well. If you talk about pathways, then some scheme or network could be useful to illustrate the point what could go wrong with learning process in your flies. I see that raw sequencing data are deposited in NCBI. Usually it is a good move to provide mapped read counts with sample information to NCBI GEO. I mean data used for the differential gene expression analysis. I briefly checked the references list. It is pretty long. maybe too long. Maybe authors could check the list again and skip some trivial references like for instance ref to the ENSEMBL consortium. Everybody knows what it is. I also could not find some recent publications, for instance PMID: 30476352 and PMID: 33608552. Although I admit it could be out of the contest of the study. And there are smaller points, which can be discussed later after making another round of a text cleaning and streamlining. Please do not fall into the description up to the very last detail. For instance exact numbers of genes are not important. If you change the test in DESeq2 line (DESeq()), then the numbers will likely change. The same will happen if you change prefiltering or switch to the edgeR protocol. By the way it is not a useless exercise to check consistency of you DEG lists at different data processing conditions.
Comments on the Quality of English LanguageEnglish is improved, but it has to be improved further. It is not only english, but also the story line. Try to be more clear and focused on the major lines. Headers and notes to the tables and figures need further improvement too.
Author Response
Comments 1: The paper is more readable than before, but some parts are still difficult to follow. I see that authors have taken suggestions and improved their tables and figures. However, headers to the tables need further improvements (table 1 for instance).
Reply 1: Table 1 and its caption were simipified.
Comments 2: Figure 3 is still too busy an barely informative.
Reply 2: We have omitted Figure 3 (the number of DEGs for TEs) from the article, only keeping a brief description of the results in the text.
Comments 3: Try to focus on the major message and provide all details. Note, that GO titles say nothing about underlying genes and gene numbers. It is important to show the gene names as well.
Reply 3: In GO and KEGG tables, we have replaced gene IDs with their names and discussed their putative functions in the text.
Comments 4: If you talk about pathways, then some scheme or network could be useful to illustrate the point what could go wrong with learning process in your flies.
Reply 4: Untfortunately, it was hard to draw a direct cause-and-effect relationship between activity of some specific pathway and Drosophila memory processes. In many cases, only a few genes for a given pathways were shown to be up- or downregulated. Putative relationships between these genes and Drosophila biological and cognitive processes are discussed in Manuscript.
Comments 5:
I see that raw sequencing data are deposited in NCBI. Usually it is a good move to provide mapped read counts with sample information to NCBI GEO. I mean data used for the differential gene expression analysis.
Reply 5: Along with raw read sequences, we have additionally provided files that contain transcriptomic reads mapped to the genome (SRA database, p. 577, PRJNA1152604). In fact, all these files can be easily obtained from the raw reads using specific commands (see Text S1).
Comments 6:
I briefly checked the references list. It is pretty long. maybe too long. Maybe authors could check the list again and skip some trivial references like for instance ref to the ENSEMBL consortium. Everybody knows what it is. I also could not find some recent publications, for instance PMID: 30476352 and PMID: 33608552. Although I admit it could be out of the contest of the study.
Reply 6:
We have omitted the ENSEMBL reference. We also added and discussed the references suggested by Reviewer, along with an additional reference regarding the influence of HMC on gene transcription profile (P. 373 – 380).
Comments 7.
And there are smaller points, which can be discussed later after making another round of a text cleaning and streamlining. Please do not fall into the description up to the very last detail. For instance exact numbers of genes are not important. If you change the test in DESeq2 line (DESeq()), then the numbers will likely change. The same will happen if you change prefiltering or switch to the edgeR protocol. By the way it is not a useless exercise to check consistency of you DEG lists at different data processing conditions.
Reply 7. We have omitted some numbers of genes in the text. Of course, the data depends on the methods of analysis. For example, we preformed DEGs analysis both with and without Wald test. The results appeared to be largely similar, especially for DEGs with the greatest degree of expression change. However, the list of DEGs without Wald test was about twice as long, and the reliability of the data was lower. We omitted Wald test in some cases (Canton-S, hypomagnetic conditions, see S1 Table). We also used both KEGG and GO analysis of enrichment, which gave largely similar data, but with some differences (see gene names and enriched pathways in KEGG and GO Tables).
Comments 8: Comments on the Quality of English Language English is improved, but it has to be improved further. It is not only english, but also the story line. Try to be more clear and focused on the major lines. Headers and notes to the tables and figures need further improvement too.
Reply 8: We have checked the manuscript, simplified the text and corrected errors.
Round 3
Reviewer 2 Report
Comments and Suggestions for Authors
The manuscript is further improved, although the title to the table 1 needs further correction. Also, fig.2. is still too vague to be appreciated, and there other problems which need fixing. I still do not understand the author's obsession with exact numbers of mutations and differentially expressed genes. It is always approximate and it changes upon shifting filtering scheme and statistical thresholds
I would accept it provided that authors make further improvement in the details.
Comments on the Quality of English LanguageEnglish is improved, although some phrases, also in Abstract section need further correction.
Author Response
Comments 1: The title to the table 1 needs further correction.
Response 1: We have changed the title of Table 1 to “Whole-genome sequences of CS and agnts3: the most important mutations”. We also made the captions under the table clearer and removed the columns containing information about the total number of changes and variant rates.
Comments 2: fig.2. is still too vague to be appreciated.
Response 2: We have сhanged Figure 2, making it clearer and more visual.
Comments 3: I still do not understand the author's obsession with exact numbers of mutations and differentially expressed genes. It is always approximate and it changes upon shifting filtering scheme and statistical thresholds.
Response 3: We believe that reporting the number of mutations and DEGs detected is standard procedure. (For example, see PMID 22728672 and PMID 24777382.) In our opinion, it helps to understand how different two strains are from each other when information about a specific assay is provided. Therefore, in the revised manuscript we emphasized that the exact number of changes depends on the procedure of analysis, for example: “In CS, our analysis method identified 718 changes and 418 strain-specific changes of the High group”. (P. 120 – 121). We have partially omitted such information from the text, where it seemed superfluous.
Comments 4: English is improved, although some phrases, also in Abstract section need further correction.
Response 4: We have corrected the errors found in the abstract, as well as some errors in other parts of the text.